# THEORETICAL BOUNDS FOR STABLE IN-CONTEXT LEARNING

## ABSTRACT

In-context learning (ICL) is flexible but its reliability is highly sensitive to prompt length. This paper establishes a non-asymptotic lower bound that links the minimal number of demonstrations to ICL stability under fixed high-dimensional sub-Gaussian representations. The bound gives explicit sufficient conditions in terms of spectral properties of the covariance, providing a computable criterion for practice. Building on this analysis, we propose a two-stage observable estimator with a one-shot calibration that produces practitioner-ready prompt-length estimates without distributional priors. Experiments across diverse datasets, encoders, and generators show close alignment between the predicted thresholds and empirical knee-points, with the theory acting as a conservative but reliable upper bound; the calibrated variant further tightens this gap. These results connect spectral coverage to stable ICL, bridge theory and deployment, and improve the interpretability and reliability of large-scale prompting in realistic finite-sample regimes.

## 1 INTRODUCTION

In-Context Learning (ICL) has become a central capability of large language models, enabling them to solve novel tasks at test time without parameter updates. Despite its flexibility, the reliability of ICL is highly sensitive to the number of demonstrations provided in the prompt. Practitioners report that the stability threshold varies widely across tasks and models, and naive scaling rules often fail to predict it accurately (Chen et al., 2023; Zhuang et al., 2020). This motivates a principled question: what is the minimal number of demonstrations required for stable ICL?

Recent theory provides partial answers. PAC-style analyses study when ICL is learnable and give finite-sample guarantees under mixture assumptions (Wies et al., 2023). Algorithmic perspectives view transformers as learners with task-dependent sample complexity, yielding upper bounds rather than minimal requirements (Li et al., 2023). Asymptotic learning curves characterize phase transitions in analytically tractable models (Lu et al., 2025). Complementary lines of work improve calibration in ICL (Zhao et al., 2021; Zhou et al., 2023; Zou et al., 2024), or reveal mechanistic insights into emergent behavior (Cho et al., 2024; Zheng et al., 2024; Wu et al., 2022). However, none of these studies provide a non-asymptotic, computable *lower bound* that directly links prompt length to stability via spectral properties (Merlevède et al., 2009; Paulin, 2015).

In this work, we address this gap by theoretically analyzing the relationship between the number of demonstrations in a prompt and the stability of ICL. Assuming the model relies on fixed high-dimensional representations, we derive an explicit non-asymptotic lower bound that characterizes when ICL remains stable with high probability. Building on this analysis, we propose a two-stage observable method with one-shot calibration that operationalizes the result in practice without requiring distributional priors (Wu et al., 2023).

Our contributions are threefold:

- We provide a non-asymptotic theoretical lower bound that, for the first time, establishes a quantitative relationship between the number of in-context demonstrations and the stability of ICL (Wies et al., 2023).

- We design an observable two-stage algorithm that estimates the required number of demonstrations in practical tasks while preserving theoretical guarantees (Li et al., 2023; Wu et al., 2023).

- We conduct extensive experiments to validate the consistency between our theoretical predictions and empirical model behavior (Chen et al., 2023; Zhuang et al., 2020).

## 2 RELATED WORKS

**Theoretical analyses.** Prior studies have examined ICL from different theoretical angles. PAC-style formulations establish when ICL is learnable under mixture models (Wies et al., 2023). Transformers have also been analyzed as algorithms with task-dependent sample complexity (Li et al., 2023), while asymptotic learning curves describe phase transitions in analytically tractable settings (Lu et al., 2025). Task-specific results for regression and structured prediction further refine these analyses (Abernethy et al., 2024).

**Calibration and reliability.** Calibration-based methods improve few-shot prediction confidence and robustness (Zhao et al., 2021; Zhou et al., 2023; Zou et al., 2024). These approaches are orthogonal to our goal: they refine predictive distributions but do not specify the minimal prompt length for stability (Zhuang et al., 2020).

**Mechanistic and empirical perspectives.** Mechanistic studies connect internal representations to emergent ICL behavior (Cho et al., 2024; Zheng et al., 2024). Empirical benchmarks and toolkits provide standardized pipelines and evaluation practices (Wu et al., 2023; Xie et al., 2021; Wu et al., 2022). Independent work observes that empirical knee-points often appear at substantially fewer examples than naive rules suggest (Chen et al., 2023), a phenomenon our theory explains through spectral coverage (Xu et al., 2023; Wu et al., 2022).

Overall, our analysis complements these lines by supplying a *lower-bound perspective* that is both non-asymptotic and computable (Tropp, 2011; Merlevède et al., 2009).

## 3 SETUP AND NOTATION (RE-STATED FOR CONVENIENCE)

**Assumptions** We analyze in-context learning under the following layered assumptions on the feature representation $\phi : \mathcal{X} \to \mathbb{R}^d$.

- **(A1) Nominal assumption (strong).** A deterministic feature map $\phi(x)$ is fixed and each $\phi(x)$ is sub-Gaussian with zero mean and covariance $\Sigma$. This assumption is adopted for the main theorem, under which all proofs are rigorous.

- **(A2) Approximate stationarity.** In practice, the representation may depend on context index $t$. We model $\phi_t(x) = \phi(x) + \Delta_t(x)$ with $\|\Delta_t(x)\|_2 \leq \varepsilon$ (uniformly in $x, t$ or in expectation). When $\varepsilon$ is small, our lower bound degrades smoothly by an additive correction term $g(\varepsilon)$ (see Appendix B).

- **(A3) Moment relaxation.** Instead of sub-Gaussian tails, we allow $\phi(x)$ to be sub-exponential or to satisfy bounded fourth-moment conditions. Concentration results can be adapted with modified constants, leading to slightly larger sample-size requirements.

- **(A4) Weak dependence.** When the context induces dependence among features $\{\phi_t(x)\}$, we assume a mixing or martingale-difference condition. In this case, matrix concentration inequalities for dependent sequences apply, yielding bounds of the same form up to a dependence factor (Appendix C).

All formal proofs in the main text are carried out under the nominal assumption (A1). The relaxed settings (A2)–(A4) are analyzed in Appendix D, where we show that the main results remain valid up to controlled perturbation terms.

- A frozen feature map $\phi : \mathcal{X} \to \mathbb{R}^d$ is assumed, satisfying $\mathbb{E}_{x \sim \mathcal{D}} \phi(x) = 0$ and $\|\phi(x)\|_{\psi_2} \leq \sigma$ (the vector $\psi_2$-Gaussian norm).

- The covariance is $\Sigma = \mathbb{E}[\phi\phi^\top] \in \mathbb{R}^{d \times d}$, with rank $r = \mathrm{rank}(\Sigma)$ (if $r < d$ we discuss on the support subspace). Denote the smallest nonzero eigenvalue by $\lambda_r(\Sigma) > 0$ and for brevity write $\lambda_{\min}(\Sigma) = \lambda_r(\Sigma)$.

- A demonstration sequence $S_K = \{x_k\}_{k=1}^K$ consists of i.i.d. draws from $\mathcal{D}$. Define

$$H_K = \begin{bmatrix} \phi(x_1) \\ \vdots \\ \phi(x_K) \end{bmatrix} \in \mathbb{R}^{K \times d}, \qquad \widehat{\Sigma}_K = \frac{1}{K} H_K H_K^\top = \frac{1}{K} \sum_{k=1}^K X_k,$$

where $X_k = \phi(x_k)\phi(x_k)^\top$ is a PSD matrix.

- Goal: give the minimal $K$ (parameterized by $\Sigma, \sigma, r$, or provide an observable two-stage estimator) such that

$$\Pr\left(\lambda_{\min}(\widehat{\Sigma}_K) \geq \delta\right) \geq 1 - \xi,$$

where $0 < \delta < \lambda_{\min}(\Sigma)$ and $\xi \in (0, 1)$.

### 3.1 BASIC MATRIX TAIL BOUND (A ONE-SIDED MATRIX BERNSTEIN FOR THE MINIMUM EIGENVALUE)

Define centered matrices $Y_k := X_k - \Sigma$, so $\mathbb{E} Y_k = 0$ and

$$\widehat{\Sigma}_K - \Sigma = \frac{1}{K} \sum_{k=1}^K Y_k.$$

We need to control $\lambda_{\min}(\widehat{\Sigma}_K) = \lambda_{\min}\left(\Sigma + \frac{1}{K} \sum_{k=1}^K Y_k\right)$. By a matrix Bernstein inequality (or a lower-tail form of Tropp-style matrix concentration), there exist constants $c, C > 0$ such that for any $t \geq 0$,

$$\Pr\left(\lambda_{\min}\left(\sum_{k=1}^K Y_k\right) \leq -t\right) \leq r_{\mathrm{eff}} \exp\left(-\frac{t^2}{2v + \frac{2}{3}Rt}\right), \tag{1}$$

where

$$v = \left\|\sum_{k=1}^K \mathbb{E}[Y_k^2]\right\| = K \cdot \left\|\mathbb{E}[(X - \Sigma)^2]\right\|$$

and $R$ is (on a high-probability event) an upper bound for $\max_k \|Y_k\|$.

Set $t = K\Delta$ where $\Delta := \lambda_{\min}(\Sigma) - \delta > 0$. Substituting into equation 1 and adjusting scaling yields for any $K$,

$$\Pr\left(\lambda_{\min}(\widehat{\Sigma}_K) \leq \delta\right) \leq r_{\mathrm{eff}} \exp\left(-\frac{K\Delta^2}{2\widetilde{v} + \frac{2}{3}R\Delta}\right), \tag{2}$$

where $\widetilde{v} := \left\|\mathbb{E}[(X - \Sigma)^2]\right\|$; the detailed derivation is in the appendix.

## 3.2 CONTROLLING PARAMETERS UNDER THE SUB-GAUSSIAN ASSUMPTION (TRUNCATION AND FOURTH-MOMENT BOUNDS)

### 3.2.1 FOURTH-MOMENT BOUND

From the vector $\psi_2$ assumption $\|\phi\|_{\psi_2} \leq \sigma$ there exists an absolute constant $C_1$ such that

$$\mathbb{E}\|\phi\|^4 \leq C_1\sigma^4. \tag{3}$$

Hence

$$\mathbb{E}\|X\|^2 = \mathbb{E}\|\phi\phi^\top\|^2 = \mathbb{E}\|\phi\|^4 \leq C_1\sigma^4. \tag{4}$$

Using the operator inequality $(X - \Sigma)^2 \preceq 2X^2 + 2\Sigma^2$ we get

$$\widetilde{v} = \left\|\mathbb{E}[(X - \Sigma)^2]\right\| \leq 2\mathbb{E}\|X\|^2 + 2\|\Sigma\|^2 \leq 2C_1\sigma^4 + 2\|\Sigma\|^2. \tag{5}$$

### 3.2.2 TRUNCATION TO CONTROL R

Typically $\|X_k\| = \|\phi(x_k)\|^2$ is not almost surely bounded, but by the sub-Gaussian tail (there exists $c_1 > 0$)

$$\Pr(\|\phi\| \geq t) \leq 2\exp\left(-\frac{c_1 t^2}{\sigma^2}\right).$$

Choose threshold $t_0 = C_2\sigma\sqrt{\log(8K/\xi)}$ (constant $C_2$ depends on $C_1$, with $C_1 C_2^2 = 1$). Then for a single sample

$$\Pr(\|\phi\| > t_0) \leq \frac{\xi}{4K}.$$

By a union bound over the $K$ samples, with probability at least $1 - \xi/4$,

$$\max_{1 \leq k \leq K}\|X_k\| = \max_k\|\phi(x_k)\|^2 \leq B_K := C_2^2\sigma^2\log\frac{8K}{\xi}.$$

On this high-probability event we may take

$$R = \|X_k - \Sigma\| \leq B_K + \|\Sigma\| \leq C_2^2\sigma^2\log\frac{8K}{\xi} + \|\Sigma\|. \tag{6}$$

## 3.3 SUBSTITUTE INTO EQUATION 2 TO OBTAIN A SAMPLE-SIZE LOWER BOUND (ONE MAIN RESULT)

Substituting equation 5 and equation 6 into equation 2, and treating the rank-related prefactor $r_{\text{eff}}$ as a logarithmic factor, require the RHS to be $\leq \xi - \xi/4$. One obtains a sufficient sample-size lower bound:

Let $\Delta = \lambda_{\min}(\Sigma) - \delta > 0$. If

$$K \geq \left[\frac{2\widetilde{v}}{\Delta^2} + \frac{2R}{3\Delta}\right]\log\frac{4r_{\text{eff}}}{3\xi}, \tag{7}$$

then (combining the truncation failure $\xi/4$ and Bernstein failure) we have

$$\Pr(\lambda_{\min}(\widehat{\Sigma}_K) \geq \delta) \geq 1 - \xi.$$

Replacing $\widetilde{v}$ and $R$ by the bounds from equation 5 and equation 6 gives the explicit bound

$$K \gtrsim \left[\frac{2(2C_1\sigma^4 + 2\|\Sigma\|^2)}{\Delta^2} + \frac{2(C_2^2\sigma^2\log(8K/\xi) + \|\Sigma\|)}{3\Delta}\right]\log\frac{4r_{\text{eff}}}{3\xi}, \tag{8}$$

where "$\gtrsim$" hides absolute constants and note $K$ appears inside the $\log(8K/\xi)$ factor (a self-consistency issue); in practice one can use a single fixed-point iteration or replace $\log(8K/\xi)$ by the conservative upper bound $\log(8K_{\max}/\xi)$.

### 3.4 A MORE CONCISE, COMMONLY USED OPERATOR-NORM CONCENTRATION (AVOID TRUNCATION AND THE SELF-CONSISTENCY)

In the higher-level literature one often uses the following operator-norm concentration (Vershynin / Koltchinskii–Lounici style): there exists a constant $C > 0$ such that for a sub-Gaussian vector

$$\Pr\left(\|\widehat{\Sigma}_K - \Sigma\| \geq C\|\Sigma\|\left(\sqrt{\frac{r_{\text{eff}} \log(2/\xi)}{K}} + \frac{r_{\text{eff}} \log(2/\xi)}{K}\right)\right) \leq \xi. \tag{9}$$

Using equation 9 to ensure $\lambda_{\min}(\widehat{\Sigma}_K) \geq \delta$ (appendix explains why this suffices) we require

$$C\|\Sigma\|\left(\sqrt{\frac{r_{\text{eff}} \log(2/\xi)}{K}} + \frac{r_{\text{eff}} \log(2/\xi)}{K}\right) \leq \Delta.$$

Ignoring the lower-order $\frac{r_{\text{eff}} \log(2/\xi)}{K}$ term (negligible when $K$ is large), we get the simpler leading sample-size term:

$$K \geq C' \frac{\|\Sigma\|^2 r_{\text{eff}} \log(2/\xi)}{\Delta^2}, \tag{10}$$

where $C'$ is a constant ($C' = C^2$). This is the common practical form.

### 3.5 A TWO-STAGE OBSERVABLE (PLUG-IN) SCHEME AND FINITE-SAMPLE GUARANTEE

The bounds in equation 8 and equation 10 depend on unknowns $\Sigma, \lambda_{\min}(\Sigma), \|\Sigma\|, r_{\text{eff}}$. We now give an observable two-stage scheme and show it works in finite samples.

**Two-stage algorithm (high level)**

1. Draw $K_0$ samples (with $K_0$ chosen theoretically below), compute $\widehat{\Sigma}_{K_0}$ and calculate

$$\widehat{\lambda}_0 := \lambda_{\min}(\widehat{\Sigma}_{K_0}), \qquad \|\Sigma\|_\wedge := \|\widehat{\Sigma}_{K_0}\|, \qquad \widehat{r}_{\text{eff}} := \frac{\operatorname{tr}(\widehat{\Sigma}_{K_0})}{\|\widehat{\Sigma}_{K_0}\|}.$$

2. Using the operator-norm deviation bound equation 9 (with $K_0$ plugged in) construct a lower-confidence bound

$$\lambda := \widehat{\lambda}_0 - C\|\Sigma\|_\wedge\left(\sqrt{\frac{\widehat{r}_{\text{eff}} \log(4/\xi)}{K_0}} + \frac{\widehat{r}_{\text{eff}} \log(4/\xi)}{K_0}\right). \tag{11}$$

Here the constant $C$ is the same as in equation 9, and the failure probability is allocated to $\xi/2$ (details below).

3. If $\lambda \leq 0$, increase $K_0$ (e.g. double it) and repeat until $\lambda > 0$. When $\lambda > 0$ set $\widehat{\Delta} := \lambda - \delta$ (if $\widehat{\Delta} \leq 0$ then $\delta$ was chosen too large), and then use equation 10 with $\|\Sigma\|$, $r_{\text{eff}}$ replaced by $\|\Sigma\|_\wedge$, $\widehat{r}_{\text{eff}}$ to compute the second-stage sample size $K$:

$$K \geq C' \frac{\|\Sigma\|_\wedge^2 \widehat{r}_{\text{eff}} \log(2/\xi)}{\widehat{\Delta}^2}.$$

4. The final total sample size is $K_0 + K$, and with total failure probability $\leq \xi$ one guarantees $\lambda_{\min}(\widehat{\Sigma}_K) \geq \delta$.

---

**Algorithm 1** Two-stage observable estimator for the minimal prompt length

---

**Require:** tolerance $\xi \in (0,1)$, target $\delta \in (0,1)$, initial $K_0$ (e.g., $K_0 = 100$)

1: **repeat**

2:    Draw $K_0$ demos, compute $\widehat{\Sigma}_{K_0}$, set $\widehat{\lambda}_0 = \lambda_{\min}(\widehat{\Sigma}_{K_0})$, $\widehat{\|\Sigma\|} = \|\widehat{\Sigma}_{K_0}\|$, $\widehat{r}_{\mathrm{eff}} = \mathrm{tr}(\widehat{\Sigma}_{K_0})/\|\widehat{\Sigma}_{K_0}\|$.

3:    Lower confidence bound (use equation 9 constant $C$):

$$\underline{\lambda} = \widehat{\lambda}_0 - C\,\widehat{\|\Sigma\|}\left( \sqrt{\frac{\widehat{r}_{\mathrm{eff}} \log(4/\xi)}{K_0}} + \frac{\widehat{r}_{\mathrm{eff}} \log(4/\xi)}{K_0} \right).$$

4:    **if** $\underline{\lambda} \le 0$ **then**

5:        $K_0 \leftarrow 2K_0$ and repeat

6:    **end if**

7: **until** $\underline{\lambda} > 0$

8: Set $\widehat{\Delta} = \underline{\lambda} - \delta$.

9: Compute the second-stage size using the plug-in form of equation 10:

$$K \;=\; C'\, \frac{\widehat{\|\Sigma\|}^2 \, \widehat{r}_{\mathrm{eff}} \log(2/\xi)}{\widehat{\Delta}^2}.$$

10: **return** total $K_{\mathrm{final}} = K_0 + K$.

---

**Finite-sample correctness (theoretical guarantee)**   Using equation 9 for the first stage: if $K_0$ satisfies

$$C\|\Sigma\|\left( \sqrt{\frac{r_{\mathrm{eff}} \log(4/\xi)}{K_0}} + \frac{r_{\mathrm{eff}} \log(4/\xi)}{K_0} \right) \le \frac{\lambda_{\min}(\Sigma)}{4}, \tag{12}$$

then with probability at least $1 - \xi/2$,

$$|\widehat{\lambda}_0 - \lambda_{\min}(\Sigma)| \le \frac{\lambda_{\min}(\Sigma)}{4}, \qquad \big| \|\widehat{\Sigma}_{K_0}\| - \|\Sigma\| \big| \le \frac{\|\Sigma\|}{4}.$$

Hence (using simple algebra $a - a/4 = 3a/4$, etc.)

$$\lambda \ge \lambda_{\min}(\Sigma) - \frac{\lambda_{\min}(\Sigma)}{4} - C \cdot \frac{5\|\Sigma\|}{4}\left( \sqrt{\frac{r_{\mathrm{eff}} \log(4/\xi)}{K_0}} + \frac{r_{\mathrm{eff}} \log(4/\xi)}{K_0} \right) \ge \frac{7\lambda_{\min}(\Sigma)}{16},$$

when $K_0$ meets equation 12. Thus there exists a constant $C''$ (e.g. $C'' = 16C^2$) such that equation 12 is equivalent to the requirement

$$K_0 \ge C'' \frac{\|\Sigma\|^2 r_{\mathrm{eff}} \log(4/\xi)}{\lambda_{\min}(\Sigma)^2}. \tag{13}$$

### 3.6   FINAL IMPLEMENTABLE THEOREM STATEMENT

**[ICL Minimal Necessary Length, non-asymptotic, observable two-stage implementation]** Under the above setup, let $\delta \in (0, \lambda_{\min}(\Sigma))$ and failure tolerance $\xi \in (0,1)$. There exist absolute constants $C_1, C_2, C_3$ (depending on matrix-Bernstein and sub-Gaussian constants) such that:

1. **(Non-observable bound using spectral quantities)** If $\Delta = \lambda_{\min}(\Sigma) - \delta > 0$ and

$$K \ge C_1\Big( \frac{\sigma^4 + \|\Sigma\|^2}{\Delta^2} + \frac{\sigma^2 \log(8K/\xi) + \|\Sigma\|}{\Delta} \Big) \log \frac{4r_{\mathrm{eff}}}{3\xi},$$

then $\Pr(\lambda_{\min}(\widehat{\Sigma}_K) \geq \delta) \geq 1 - \xi$. Ignoring lower-order terms (and using the sharper operator-norm concentration) yields the simplified dominant form

$$K \geq C_2 \frac{\|\Sigma\|^2 r_{\text{eff}} \log(2/\xi)}{(\lambda_{\min}(\Sigma) - \delta)^2},$$

which suffices in practice.

2. **(Two-stage observable implementation)** If the first-stage sample size $K_0$ satisfies

$$K_0 \geq C_3 \frac{\|\Sigma\|^2 r_{\text{eff}} \log(4/\xi)}{\lambda_{\min}(\Sigma)^2},$$

then forming $\widehat{\Sigma}_{K_0}$ and constructing the lower confidence bound $\lambda$ as in equation 11, and then setting $\widehat{\Delta} = \lambda - \delta$ and choosing $K$ according to the rule in item (1) (with plug-in estimates $\|\Sigma\|_{\wedge}$, $\widehat{r}_{\text{eff}}$) yields, with total failure probability $\leq \xi$, the guarantee $\lambda_{\min}(\widehat{\Sigma}_{K_0+K}) \geq \delta$.

## 3.7 CONNECTION TO ICL (IN-CONTEXT LEARNING) AND EXPERIMENTAL SUGGESTIONS

**A one-shot practical calibration.** To tighten $K^*$ for practice, we (i) replace $\|\Sigma\|$ by $\|\Sigma\|_{\text{F}}$, (ii) use the trace-based effective rank $r_{\text{eff}}^{\text{tr}} = \text{tr}(\Sigma)^2/\|\Sigma\|_{\text{F}}^2$, and (iii) substitute the minimum eigenvalue by a $q$-quantile eigenvalue $\lambda_q$ ($q \in [0.05, 0.2]$ tuned on a small validation set; no test leakage). We then compute

$$K^{\text{cal}} = \alpha \cdot \frac{\|\widehat{\Sigma}\|_{\text{F}}^2 \, r_{\text{eff}}^{\text{tr}} \, \log(2/\xi)}{(\lambda_q - \delta)^2},$$

where a single global scale $\alpha$ is estimated once per encoder–generator pair from a small validation subset (e.g., 5 tasks). This variant preserves the same monotone dependence as the theory while substantially reducing over-conservativeness.

**Theoretical meaning** Using $\lambda_{\min}(\widehat{\Sigma}_K)$ as a proxy for "ICL learnability/stability" is reasonable because the condition number (and hence predictive variance in ridge regression or linear approximation) depends on inverse eigenvalues. For example, prediction variance scales with $\phi(x^*)^\top (H_K^\top H_K + \lambda I)^{-1} \phi(x^*)$, which is proportional to $(K\lambda_{\min}(\widehat{\Sigma}_K) + \lambda)^{-1}$. If $\lambda_{\min}(\widehat{\Sigma}_K)$ is lower bounded by $\delta$, then prediction variance is upper bounded and ICL is more likely to behave stably.

**Empirical validation of assumptions.** Although our main theorem is derived under the nominal assumption (A1), we conduct diagnostic experiments on hidden representations from GPT-2 models to evaluate how well these assumptions hold in practice. We find that (i) hidden states are approximately sub-Gaussian at intermediate layers, though heavier tails appear in later layers; (ii) representation drift across prompt positions is present but moderate; (iii) the spectral statistic $\lambda_{\min}(\hat{\Sigma}_K)$ correlates with empirical ICL stability. These results indicate that our theoretical assumptions serve as a reasonable proxy, and the relaxed extensions in Appendix D capture the observed deviations.

**Synthetic stress tests.** To further validate our theoretical claims, we perform controlled synthetic stress tests (Appendix E) by introducing bounded drift, heavy-tailed features, and temporal dependence. Across all cases, we find that empirical knee-points increase smoothly with the violation strength, in line with our robustness extensions, confirming that the lower bound remains a conservative but reliable predictor of ICL stability.

Table 1: Error ratio ($K^*$/knee) across datasets, encoders, and generators. Each number is averaged over 5 random seeds. Lower is better (closer to 1). The Raw column uses the default parameter values for prediction, whereas the Calibrated column reflects a few exploratory adjustments to some of these values; the procedure for determining each parameter is detailed in H.2.

| Dataset | Encoder | GPT-2 small (124M) | | GPT-2 medium (345M) | | GPT-3 (175B) | |
|---|---|---|---|---|---|---|---|
| | | Raw | Calibrated | Raw | Calibrated | Raw | Calibrated |
| SST-2 | all-mpnet-base-v2 | 3.4× | 1.2× | 2.9× | 1.0× | 2.0× | 1.0× |
| | all-MiniLM-L6-v2 | 2.6× | 1.0× | 2.2× | 1.0× | 1.9× | 1.0× |
| | paraphrase-MiniLM-L12-v2 | 3.2× | 1.1× | 2.8× | 1.1× | 2.0× | 1.0× |
| AGNews | all-mpnet-base-v2 | 3.4× | 1.0× | 2.9× | 1.0× | 2.3× | 1.0× |
| | all-MiniLM-L6-v2 | 2.8× | 1.0× | 2.4× | 1.0× | 1.9× | 1.0× |
| | paraphrase-MiniLM-L12-v2 | 3.0× | 1.1× | 2.8× | 1.0× | 2.0× | 1.0× |
| Yahoo! Answers | all-mpnet-base-v2 | 3.1× | 1.1× | 2.8× | 1.1× | 1.8× | 1.0× |
| | all-MiniLM-L6-v2 | 2.6× | 1.1× | 2.3× | 1.0× | 1.6× | 1.0× |
| | paraphrase-MiniLM-L12-v2 | 2.9× | 1.0× | 2.7× | 1.0× | 1.8× | 1.0× |
| TREC-6 | all-mpnet-base-v2 | 3.3× | 1.0× | 2.8× | 1.0× | 2.1× | 1.0× |
| | all-MiniLM-L6-v2 | 2.7× | 1.0× | 2.4× | 1.0× | 2.0× | 1.0× |
| | paraphrase-MiniLM-L12-v2 | 3.2× | 1.0× | 2.8× | 1.0× | 2.0× | 1.0× |

**Remark on interpretation.** While we use $\lambda_{\min}(\hat{\Sigma}_K)$ as a stability proxy, we recognize that it can be conservative and may not fully capture all sources of instability (e.g. attention saturation). Appendix F provides robustness analyses, alternative spectral surrogates, and empirical diagnostics that validate and calibrate this proxy.

## 4 EXPERIMENTS

We conduct experiments to validate the theoretical lower bound on the minimal prompt length required for stable In-Context Learning (ICL). The experiments test the predictions of our theory across various datasets, encoders, and generators.

### 4.1 MEASURING THE KNEE-POINT

The knee-point $\widehat{K}_{\mathrm{knee}}$ is defined as the point of minimal sum of squared errors (SSE) in a two-segment piecewise linear fit of the accuracy curve $\{(K, \mathrm{ACC}(K))\}_{K \in \mathcal{K}}$. We calculate the knee-point by selecting the split index $t$ that minimizes SSE, and the uncertainty is quantified via bootstrapping with 200 resamplings, reporting the 95% percentile interval for $\widehat{K}_{\mathrm{knee}}$.

### 4.2 RESULTS ACROSS DATASETS

Table 1 summarizes the empirical knee-points across datasets. Theoretical predictions consistently exceed empirical knee-points with an average error ratio of 2.9×, confirming the reliability of the theoretical upper bound. The calibrated estimates ($K_{\mathrm{cal}}$) track the knee-point more closely.

### 4.3 TASK-DEPENDENCE OF ERROR RATIOS

The error ratio varies across datasets. More complex tasks, such as AGNews, require larger prompts compared to simpler tasks like TREC-6. This aligns with our theory, which predicts larger prompts for more complex tasks.

### 4.4 MODEL SCALE EFFECT

We evaluate the effect of model scale on ICL performance by testing a 124M and a 345M model. Larger models reduce the empirical knee-point, and the error ratio improves. For SST-2, the knee decreases from $\approx 12$ for the 124M model to $\approx 7$ for the 345M model, with the error ratio decreasing from 3.4× to 2.2×.

### 4.5 IMPACT OF TOKENIZATION AND PROMPT TRUNCATION

The choice of tokenization and prompt truncation policies significantly affects ICL stability. Single-token labels (e.g., "yes/no") lead to smaller knees and error ratios (1.2–1.5×), whereas multi-token labels (e.g., "positive/negative") result in error ratios greater than 3×. For truncation, "latest-first" truncation produces the smallest knees and error ratios (1.2–1.8×), while "earliest-first" and "random" truncations lead to larger error ratios.

### 4.6 SYNTHETIC STRESS TESTS

We further evaluate the robustness of our theoretical bound by introducing controlled violations of the nominal assumption (A1). These stress tests involve the following: **Bounded Drift:** As the drift parameter $\epsilon$ increases, the empirical knee-point increases linearly. This confirms the additive correction predicted by our robustness analysis in B. **Heavy-Tailed Features:** As the tail parameter $\nu$ decreases, the empirical knee-point increases multiplicatively. However, truncation or normalization significantly reduces this inflation, as detailed in D. **Ablation Studies** is shown in G.

These observations confirm that the lower bound remains a reliable predictor of ICL stability across different data violations.

## 5 CONCLUSION

Our experiments validate an important property: theoretical predictions reliably upper bound the empirical saturation point of In-Context Learning (ICL). This provides a practical guideline for estimating the minimal number of demonstrations required for stable ICL, avoiding excessive computational costs and long sequences. This work advances the understanding of ICL's stability by directly linking it to measurable statistical properties, offering a foundation for improving ICL performance in future work.

The theoretical framework bridges high-dimensional statistical properties and ICL behavior, showing that spectral characteristics, such as minimum eigenvalue and effective rank, influence model stability. These insights lay the groundwork for refining ICL models, particularly in real-world applications. While effective across various tasks, the current analysis assumes fixed high-dimensional representations, which may limit generalization. Future work could explore relaxing this assumption and further refining calibration procedures for more complex tasks to enhance robustness and performance.

Looking ahead, extending this approach to multi-modal ICL and integrating adaptive prompting strategies would further improve model flexibility and stability in dynamic environments, opening new avenues for practical applications of large-scale language models.

## 6 REPRODUCIBILITY STATEMENT

We ensure reproducibility by providing detailed documentation of all hyperparameters, model architectures, and experimental setups in Appendix. All datasets and architectures used in this paper are publicly available, and our main code is shown in supplementary material.

## 7 ETHICS STATEMENT

This work uses only public datasets and code under their original licenses. No private or personally identifiable information (PII) was accessed or processed. We audited the datasets for sensitive attributes when applicable and used them solely for research and benchmarking purposes. Potential negative impacts mainly arise from misuse of the trained models or analyses, e.g., deploying them in high-stakes settings without additional validation. To mitigate risks, we (i) release code and configurations for reproducibility, (ii) include usage notes that restrict deployment to research contexts, and (iii) report known failure modes and assumptions. Computation was limited to standard academic GPUs; we estimate compute/energy use is modest relative to contemporary large-scale training, and we adopted checkpoint reuse and early stopping to reduce waste. This statement is consistent with our LLM usage disclosure, which was limited to writing polish and literature discovery and did not involve sensitive data.

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

## A  USE OF LARGE LANGUAGE MODELS.

We used large language models (LLMs) in two limited ways. First, for *aid or polish writing*, an LLM service (provider redacted for anonymous review) helped with grammar, style, and tightening of sentences. All technical content, problem formulations, theorems, proofs, algorithmic designs, model implementations, experiments, and final claims were authored and verified by the authors. Second, for *retrieval and discovery*, we used LLM-assisted search to expand keyword queries and to draft short summaries of candidate papers from public abstracts. Inclusion decisions and all citations were made only after the authors read the original sources; no autogenerated references were accepted. No private or proprietary data were provided to the LLM, and prompts contained only our own drafts or publicly available text. This use does not affect reproducibility: the released code, data, and results do not depend on any LLM outputs.

## B  ROBUSTNESS EXTENSIONS

In the main text, our theoretical guarantees are derived under the nominal assumption (A1), namely that the representation $\phi(x)$ is fixed and sub-Gaussian with covariance $\Sigma$. Here we discuss how the results extend when assumptions (A2)–(A4) are considered.

### B.1  ROBUSTNESS TO REPRESENTATION DRIFT (A2)

We first allow the representation to vary with context index $t$, modeled as

$$\phi_t(x) = \phi(x) + \Delta_t(x),$$

where $\|\Delta_t(x)\|_2 \leq \varepsilon$ uniformly in $x, t$. Let $\Sigma_t = \mathbb{E}[\phi_t(x)\phi_t(x)^\top]$ and denote the averaged covariance by $\bar{\Sigma} = \frac{1}{T} \sum_{t=1}^{T} \Sigma_t$.

**Robustness under bounded drift**     Let $K_{\text{nominal}}$ be the sample-size lower bound under (A1), depending on $\Sigma$. Under (A2), the empirical covariance $\hat{\Sigma}_K$ concentrates around $\bar{\Sigma}$, and for any $\delta \in (0, \lambda_{\min}(\bar{\Sigma}))$, with probability at least $1 - \xi$ we have

$$\lambda_{\min}(\hat{\Sigma}_K) \ \geq \ \delta - c_1 \varepsilon,$$

provided

$$K \ \geq \ K_{\text{nominal}} \ + \ c_2 \cdot \frac{\varepsilon^2 \|\Sigma\|}{(\lambda_{\min}(\Sigma) - \delta)^2} \, ,$$

where $c_1, c_2$ are absolute constants.

**Proof sketch.**     Write $\phi_t(x) = \phi(x) + \Delta_t(x)$ and expand $\Sigma_t = \Sigma + \mathbb{E}[\phi(x)\Delta_t(x)^\top] + \mathbb{E}[\Delta_t(x)\phi(x)^\top] + \mathbb{E}[\Delta_t(x)\Delta_t(x)^\top]$. The last three terms are bounded in operator norm by $O(\varepsilon\|\Sigma\|^{1/2} + \varepsilon^2)$. Applying Weyl's inequality gives $\lambda_{\min}(\Sigma_t) \geq \lambda_{\min}(\Sigma) - O(\varepsilon)$. Matrix Bernstein then yields concentration of $\hat{\Sigma}_K$ around $\bar{\Sigma}$ with the same rate as before, plus an additive slack proportional to $\varepsilon$. The additional sample-size term arises from controlling the variance proxy $v$ with $\varepsilon$-dependent terms. $\qquad\square$

## B.2   RELAXING TAIL ASSUMPTIONS (A3)

If $\phi(x)$ is only sub-exponential or has bounded fourth moments, then $\|\phi(x)\|^4$ has heavier tails. In this case, truncation plus Bernstein inequality yields an adjusted bound of the form

$$K \ \gtrsim \ \frac{(\|\Sigma\|^2 + \sigma^4 + \kappa^2)}{(\lambda_{\min}(\Sigma) - \delta)^2} \, \text{polylog}(d/\xi),$$

where $\kappa$ is a moment constant. The dependence on $\kappa$ reflects the heavier tails, but the overall scaling remains quadratic in the spectral gap.

## B.3   WEAK DEPENDENCE AMONG FEATURES (A4)

When $\{\phi_t(x)\}$ are not independent across $t$, we apply matrix concentration for martingale-difference or $\alpha$-mixing sequences (e.g., Tropp 2011, Paulin 2015). In this case, the effective sample size is reduced: if the mixing coefficient is $\rho \in [0, 1)$, then the bound takes the same form as in Theorem B.1, with $K$ replaced by $K/(1 + \rho)$. Thus moderate dependence inflates the sample complexity by at most a constant factor.

## B.4   SUMMARY

Together, these results demonstrate that our main lower bound is *robust*: when the representation drifts mildly, exhibits heavier tails, or has weak temporal dependence, the sufficient sample-size requirement increases smoothly and predictably, rather than collapsing. In practice, this means that the nominal bound from (A1) can still be used as a conservative guideline, while deviations are accommodated by additive or multiplicative correction factors.

## C   EXTENSIONS TO DEPENDENT FEATURES

Our main theorem assumes that feature vectors $\{\phi_t(x)\}$ are i.i.d. across prompt positions. In practice, contextualized representations may exhibit correlations across $t$ due to attention and residual connections. We now extend the analysis under weak dependence assumptions.

Table 2: Notation and constants used across sections.

| Symbol | Meaning / Where defined |
|---|---|
| $\Sigma$, $\|\Sigma\|$ | Covariance and operator norm (Sec. 3) |
| $r_{\text{eff}} = \text{tr}(\Sigma)/\|\Sigma\|$ | Effective rank (Sec. 3; Appx. G.2) |
| $\sigma$ | Sub-Gaussian parameter (Appx. 3.2; Appx. G.3) |
| $\delta$ | Target lower bound for $\lambda_{\min}(\widehat{\Sigma}_K)$ |
| $C$ | Constant in operator-norm deviation equation(9) |
| $C', C''$ | Constants in plug-in lower bounds equation(10), (13) |
| $B_K$ | Truncation threshold (equation(6)) |
| $\lambda_q$ | $q$-quantile eigenvalue surrogate (Appx. F) |

## C.1 MARTINGALE-DIFFERENCE SETTING

Suppose $\{\phi_t(x)\}$ forms a martingale-difference sequence with respect to a natural filtration $\mathcal{F}_t$ (e.g., conditioned on past tokens). That is, $\mathbb{E}[\phi_t(x) \mid \mathcal{F}_{t-1}] = 0$. Then the sequence of covariance deviations

$$X_t = \phi_t(x)\phi_t(x)^\top - \mathbb{E}[\phi_t(x)\phi_t(x)^\top \mid \mathcal{F}_{t-1}]$$

is a matrix martingale-difference sequence. By applying the matrix Azuma–Hoeffding inequality (Tropp 2011; Hsu et al. 2012), we obtain concentration of $\hat{\Sigma}_K = \frac{1}{K}\sum_{t=1}^K \phi_t(x)\phi_t(x)^\top$ around its expectation, with variance proxy and tail probability essentially the same as in the i.i.d. case, up to a constant factor.

**Lower bound under martingale differences** Under Assumption (A4) with bounded martingale differences, the non-asymptotic lower bound on $K$ in Theorem 3.1 remains valid up to replacing $\xi$ by $c \cdot \xi$ for an absolute constant $c > 0$.

## C.2 MIXING SEQUENCES

Alternatively, suppose $\{\phi_t(x)\}$ is $\alpha$-mixing with coefficient $\alpha(s) \leq \rho^s$ for some $\rho \in (0,1)$. Following Paulin (2015) and Merlevède et al. (2009), matrix Bernstein inequalities for mixing sequences yield concentration with an effective sample size

$$K_{\text{eff}} \approx \frac{K}{1 + c_\rho},$$

where $c_\rho = \sum_{s \geq 1} \alpha(s)$ quantifies temporal dependence. In particular, geometric mixing ($\alpha(s) \leq \rho^s$) gives $c_\rho = O(1/(1-\rho))$, so that moderate dependence inflates the required prompt length by at most a constant factor.

**Lower bound under mixing** Under Assumption (A4) with mixing coefficient $\rho < 1$, the sufficient condition on prompt length becomes

$$K \gtrsim (1 + c_\rho) \cdot \frac{\sigma^2 \|\Sigma\|}{(\lambda_{\min}(\Sigma) - \delta)^2} \log\frac{d}{\xi}.$$

## C.3 DISCUSSION

These results show that our main conclusions are *robust to weak dependence*. Even if contextualized features $\{\phi_t(x)\}$ are not independent, as long as dependence decays over time (mixing) or is conditionally centered

(martingale-difference), the sufficient sample size bound degrades only by a multiplicative factor, rather than failing altogether. In practice, this suggests that $\lambda_{\min}(\hat{\Sigma}_K)$ can still serve as a useful stability proxy, with dependence merely shifting the knee-point moderately to the right.

# D    EXTENSIONS BEYOND THE NOMINAL ASSUMPTION

In the main paper we derived our non-asymptotic lower bound under the nominal assumption (A1), namely that representations $\phi(x)$ are fixed, i.i.d., and sub-Gaussian with covariance $\Sigma$. Here we provide extensions under weaker assumptions (A2)–(A4), complementing Appendix B and Appendix C. Overall, these results show that our theory is *robust*: the sufficient prompt length bound degrades smoothly under moderate violations of (A1), rather than failing abruptly.

## D.1    REPRESENTATION DRIFT (A2)

As discussed in Appendix B, if the representation depends on the context index $t$, modeled as $\phi_t(x) = \phi(x) + \Delta_t(x)$ with $\|\Delta_t(x)\|_2 \leq \varepsilon$, then eigenvalue perturbation bounds (Weyl's inequality) imply that $\lambda_{\min}(\Sigma_t)$ is reduced by at most $O(\varepsilon)$. Combining this with matrix Bernstein yields a lower bound of the form

$$K \ \geq \ K_{\mathrm{nominal}} + g(\varepsilon),$$

with $g(\varepsilon)$ polynomial in $\varepsilon$ and spectral ratios. Hence, small drift only shifts the required prompt length slightly.

## D.2    RELAXED TAIL ASSUMPTIONS (A3)

If $\phi(x)$ has heavier tails than sub-Gaussian—e.g., only sub-exponential or bounded fourth moments—then truncation and Bernstein's inequality (Vershynin, 2018) give concentration at a slightly slower rate. In this case, the sufficient prompt length becomes

$$K \ \gtrsim \ \frac{(\|\Sigma\|^2 + \kappa^2)}{(\lambda_{\min}(\Sigma) - \delta)^2}\, \mathrm{polylog}(d/\xi),$$

where $\kappa$ is a tail parameter depending on the fourth moment or sub-exponential constant. Thus, the order of the bound remains unchanged, but the constant is inflated.

## D.3    WEAK DEPENDENCE (A4)

When features are not independent across $t$, we can use concentration for martingale-difference sequences (Tropp, 2011) or $\alpha$-mixing processes (Paulin, 2015; Merlevède et al., 2009). Both cases yield an effective sample size of the form $K_{\mathrm{eff}} = K/(1 + c_\rho)$, where $c_\rho$ is a dependence factor. Thus, weak dependence inflates the sufficient prompt length by a multiplicative constant, but does not alter the overall scaling.

## D.4    SUMMARY OF EXTENSIONS

The results under (A2)–(A4) can be summarized as follows:

- Drift in representations adds an additive correction $g(\varepsilon)$, linear or quadratic in the drift magnitude.
- Heavy-tailed features require an inflated constant, governed by moment parameters, but retain the same quadratic dependence on the spectral gap.

- Temporal dependence reduces the effective sample size, inflating the required $K$ by a constant factor depending on mixing strength.

Taken together, these extensions demonstrate that the main non-asymptotic bound is conservative but stable: it remains meaningful even when real-world deviations from (A1) occur, providing a useful guideline for interpreting empirical ICL stability.

Table 3: Summary of assumptions and their effect on the sufficient prompt length $K$. Here $K_{\text{nominal}}$ denotes the bound under (A1).

| Assumption | Description | Effect on lower bound |
|---|---|---|
| (A1) Nominal | Fixed sub-Gaussian features with covariance $\Sigma$ | $K \geq K_{\text{nominal}} = O\left(\frac{\sigma^2 \|\Sigma\|}{(\lambda_{\min}(\Sigma) - \delta)^2} \log \frac{d}{\xi}\right)$ |
| (A2) Drift | Context-dependent features $\phi_t(x) = \phi(x) + \Delta_t(x)$, with $\|\Delta_t(x)\| \leq \varepsilon$ | $K \geq K_{\text{nominal}} + g(\varepsilon)$, where $g(\varepsilon)$ grows polynomially in $\varepsilon$ and spectral ratios |
| (A3) Heavy tails | Sub-exponential or bounded fourth moments, parameter $\kappa$ | $K \gtrsim \frac{\|\Sigma\|^2 + \kappa^2}{(\lambda_{\min}(\Sigma) - \delta)^2} \operatorname{polylog}(d/\xi)$ |
| (A4) Dependence | Features form a martingale-difference sequence or $\alpha$-mixing process with coefficient $\rho$ | Effective sample size $K_{\text{eff}} \approx \frac{K}{1 + c_\rho}$, so bound inflates by factor $(1 + c_\rho)$ |

Table 4: Diagnostics of representation assumptions on GPT-2 small. Reported values are averages across 500 prompts with 95% bootstrap confidence intervals in parentheses.

| Prompt length $K$ | Tail kurtosis | Drift $D(K)$ | $\lambda_{\min}(\hat{\Sigma}_K)$ | ICL acc. (%) |
|---|---|---|---|---|
| 16 | 3.1 (2.9–3.4) | 0.05 | 0.42 (0.40–0.44) | 58.2 |
| 32 | 3.3 (3.0–3.6) | 0.08 | 0.38 (0.36–0.40) | 61.7 |
| 64 | 3.8 (3.5–4.2) | 0.12 | 0.35 (0.33–0.37) | 65.9 |
| 128 | 4.5 (4.1–4.9) | 0.21 | 0.28 (0.26–0.30) | 68.1 |

**Interpretation.** The empirical kurtosis values are close to the Gaussian baseline (3.0) for short prompts but increase with $K$, suggesting heavier tails at longer contexts, consistent with Assumption (A3). The drift statistic $D(K)$ grows gradually, supporting Assumption (A2) with small but non-negligible $\varepsilon$. The spectral statistic $\lambda_{\min}(\hat{\Sigma}_K)$ decreases as $K$ increases, and its trend correlates with the empirical knee-point in ICL accuracy. Bootstrap confidence intervals confirm that these trends are statistically significant. Overall, the diagnostics support our claim that the nominal assumption (A1) is a reasonable approximation in practice, while the relaxed extensions (A2)–(A4) capture the observed deviations.

# E    Synthetic Stress Tests

To evaluate the robustness of our theoretical bound under controlled violations of Assumption (A1), we conducted synthetic stress tests along three axes: (i) bounded drift in representations (A2), (ii) heavy-tailed features (A3), and (iii) temporal dependence (A4). For each setting we generated synthetic features and tasks as described in Section G.8, measured empirical knee-points $K_{\text{emp}}$ of in-context learning performance, and compared them against the theoretical lower bound $K_{\text{nominal}}$.

### E.1 BOUNDED DRIFT

Table 5: Drift stress test results ($d = 50$). $K_{\text{nominal}} = 32$ computed by equation 10. $K_{\text{emp}}$ is reported with 95% bootstrap CI.

| $\varepsilon$ | $K_{\text{nominal}}$ | $K_{\text{emp}}$ | $\Delta K$ |
|---|---|---|---|
| 0.00 | 32 | $30 \pm 3$ | -2 |
| 0.01 | 32 | $35 \pm 4$ | 3 |
| 0.05 | 32 | $48 \pm 6$ | 16 |
| 0.10 | 32 | $70 \pm 8$ | 38 |

**Observation.** Empirical knee-points increase smoothly with $\varepsilon$, and $\Delta K$ is well fitted by a linear function in $\varepsilon$, confirming the additive correction $g(\varepsilon)$ predicted by our robustness theorem.

### E.2 HEAVY-TAILED FEATURES

Table 6: Heavy-tail stress test results ($d = 200$). Lower $\nu$ corresponds to heavier tails. $K_{\text{nominal}} = 64$.

| $\nu$ | $K_{\text{nominal}}$ | $K_{\text{emp}}$ | $\Delta K$ |
|---|---|---|---|
| $\infty$ (Gaussian) | 64 | $62 \pm 4$ | -2 |
| 10 | 64 | $70 \pm 5$ | 6 |
| 5 | 64 | $92 \pm 7$ | 28 |
| 3 | 64 | $120 \pm 9$ | 56 |

**Observation.** Heavier tails inflate the required prompt length by a multiplicative constant, but truncation or normalization substantially reduces the inflation. This behavior aligns with the extended bound in Appendix D.

### E.3 TEMPORAL DEPENDENCE

Table 7: Dependence stress test results ($d = 100$). $K_{\text{nominal}} = 32$. Effective sample size $K_{\text{eff}}$ decreases with $\rho$.

| $\rho$ | $K_{\text{nominal}}$ | $K_{\text{emp}}$ | $\Delta K$ | $K_{\text{eff}}$ est. |
|---|---|---|---|---|
| 0.0 | 32 | $30 \pm 3$ | -2 | 32 |
| 0.2 | 32 | $38 \pm 4$ | 6 | 27 |
| 0.5 | 32 | $50 \pm 6$ | 18 | 21 |
| 0.8 | 32 | $70 \pm 8$ | 38 | 16 |

**Observation.** Empirical knee-points scale approximately as $(1 + c_\rho) \cdot K_{\text{nominal}}$, consistent with our dependence extension in Appendix C.

### E.4 SUMMARY

Across all three stress tests, the empirical knee-point increases smoothly with the violation strength, either additively (drift), multiplicatively (tails), or by a dependence factor (temporal correlation). These controlled

experiments confirm that our non-asymptotic lower bound is robust: violations of (A1) shift the threshold conservatively, but do not invalidate the predictive power of $\lambda_{\min}(\Sigma)$ as a proxy for ICL stability.

## F    INTERPRETABILITY OF THE PROXY

**Predictive variance control via $\lambda_{\min}$**    Let $\Phi \in \mathbb{R}^{K \times d}$ be the design with rows $\phi(x_k)^\top$, $\widehat{\Sigma}_K = \frac{1}{K}\Phi^\top\Phi$. Assume $y_k = f^\star(x_k) + \eta_k$, $\eta_k \overset{\text{i.i.d.}}{\sim} \mathcal{N}(0,\sigma^2)$, and $\|\phi(x)\|_2 \leq B$. For ridge predictor $\widehat{f}(x) = \phi(x)^\top \left(\widehat{\Sigma}_K + \frac{\lambda}{K}I\right)^{-1} \frac{1}{K}\Phi^\top y$, if $\lambda_{\min}(\widehat{\Sigma}_K) \geq \delta > 0$, then

$$\mathrm{Var}\left(\widehat{f}(x_\star) \mid \Phi\right) \leq \frac{\sigma^2 B^2}{K\,\delta + \lambda}.$$

*Proof.* See F.1 for details; it follows by the matrix inequality $A^{-1}(K\widehat{\Sigma}_K)A^{-1} \preceq A^{-1}$ with $A = K\widehat{\Sigma}_K + \lambda I$ and the Rayleigh quotient bound $\mathbf{v}^\top A^{-1}\mathbf{v} \leq \|\mathbf{v}\|_2^2/\lambda_{\min}(A)$. $\qquad\square$

**Theoretical link.**    Proposition F (Appendix F) shows that a lower bound on $\lambda_{\min}(\hat{\Sigma}_K)$ suffices to control predictive variance of in-context regression, providing a mechanistic rationale for why it correlates with stability.

**Limitations.**    However, $\lambda_{\min}$ can be overly conservative in practice: (i) tiny eigenvalues with negligible test alignment, (ii) heavy tails beyond sub-Gaussian assumptions, (iii) contextual dependence that alters effective conditioning.

**Alternative surrogates.**    We therefore also consider quantile eigenvalues $\lambda_q$, the condition number $\kappa$, effective rank $r_{\text{eff}}$, and average leverage scores. These quantities often correlate more closely with empirical knee-points.

**Calibration.**    A light calibration step (estimating a global scale $\alpha$ and quantile level $q$ on a small validation set) substantially improves predictive accuracy of the proxy (see Table 1).

### F.1    PROOF SKETCH OF PROPOSITION

**Setup.**    Let $\Phi \in \mathbb{R}^{K \times d}$ be the design matrix with rows $\phi(x_k)^\top$, and denote $\hat{\Sigma}_K := \frac{1}{K}\Phi^\top\Phi$. Assume the response obeys $y_k = f^\star(x_k) + \eta_k$ with homoscedastic noise $\eta_k \overset{\text{i.i.d.}}{\sim} \mathcal{N}(0,\sigma^2)$, and $\|\phi(x)\|_2 \leq B$ almost surely. Consider the ridge predictor

$$\hat{f}(x) = \phi(x)^\top \left(\hat{\Sigma}_K + \frac{\lambda}{K}I\right)^{-1} \cdot \left(\frac{1}{K}\Phi^\top y\right), \quad \lambda \geq 0.$$

For a fixed test point $x^\star$, write $\phi^\star := \phi(x^\star)$.

**Linear form and conditional variance.**    Conditioned on the design $\Phi$, $\hat{f}(x^\star)$ is linear in $y$:

$$\hat{f}(x^\star) = a^\top y, \qquad a^\top = \phi^{\star\top} \left(\hat{\Sigma}_K + \frac{\lambda}{K}I\right)^{-1} \frac{1}{K}\Phi^\top.$$

Since $\mathrm{Cov}(y \mid \Phi) = \sigma^2 I_K$, we obtain

$$\mathrm{Var}\left(\hat{f}(x^\star) \mid \Phi\right) = \sigma^2 \|a\|_2^2 = \sigma^2 \phi^{\star\top} \left(\hat{\Sigma}_K + \frac{\lambda}{K}I\right)^{-1} \hat{\Sigma}_K \left(\hat{\Sigma}_K + \frac{\lambda}{K}I\right)^{-1} \phi^\star.$$

Equivalently, with $A := K\hat{\Sigma}_K + \lambda I$,

$$\text{Var}\Big(\hat{f}(x^\star) \mid \Phi\Big) = \sigma^2\, \phi^{\star\top} A^{-1}\, (K\hat{\Sigma}_K)\, A^{-1} \phi^\star. \tag{14}$$

**Step 1: matrix inequality.** Note that $0 \preceq K\hat{\Sigma}_K \preceq A = K\hat{\Sigma}_K + \lambda I$. Therefore $A^{-1/2}(K\hat{\Sigma}_K)A^{-1/2} \preceq I$, which implies

$$A^{-1}(K\hat{\Sigma}_K)A^{-1} \preceq A^{-1}.$$

Plugging into equation 14 yields

$$\text{Var}\Big(\hat{f}(x^\star) \mid \Phi\Big) \leq \sigma^2\, \phi^{\star\top} A^{-1} \phi^\star. \tag{15}$$

**Step 2: Rayleigh quotient bound.** For any $M \succ 0$ and vector $v$, $v^\top M^{-1} v \leq \|v\|_2^2/\lambda_{\min}(M)$. Taking $M = K\hat{\Sigma}_K + \lambda I$ and $v = \phi^\star$, and using $\|\phi^\star\|_2 \leq B$, we obtain

$$\phi^{\star\top} A^{-1} \phi^\star \leq \frac{\|\phi^\star\|_2^2}{K\,\lambda_{\min}(\hat{\Sigma}_K) + \lambda} \leq \frac{B^2}{K\,\lambda_{\min}(\hat{\Sigma}_K) + \lambda}.$$

Combining with equation 15,

$$\text{Var}\Big(\hat{f}(x^\star) \mid \Phi\Big) \leq \frac{\sigma^2 B^2}{K\,\lambda_{\min}(\hat{\Sigma}_K) + \lambda}.$$

**Conclusion.** Thus, whenever $\lambda_{\min}(\hat{\Sigma}_K) \geq \delta$,

$$\text{Var}(\hat{f}(x^\star)) \leq \frac{\sigma^2 B^2}{K\delta + \lambda},$$

which establishes Proposition F. $\qquad\square$

**Remarks.** (i) If $\lambda = 0$, the result still holds provided $\hat{\Sigma}_K \succ 0$. (ii) For heteroscedastic noise, the bound uses the maximal variance. (iii) Under model misspecification, bias terms can be separated, and the variance part remains controlled by $\lambda_{\min}(\hat{\Sigma}_K)$.

## G ABLATION STUDIES

**Overview.** To bridge the gap between theory and practice, we performed ten ablations that correspond to the key factors appearing in our formula and experimental pipeline. We first present a summary table that aggregates all results, followed by ten dedicated \subsections, each restating its own result with brief analysis and takeaway. All numbers are averages over multiple runs unless otherwise noted, and error ratios are defined as theoretical $K^*$ divided by the empirical knee (lower is better).

### G.1 RATIO SENSITIVITY

**Result.** Increasing $\delta$ from $0.1$ to $0.9$ reduces the theoretical upper bound from $K^* \approx 95$ to $K^* \approx 25$, with error ratio decreasing from $5.3\times$ to $1.4\times$. The mechanism is through the effective spectral gap $\Delta = \lambda_{\min}(\Sigma) - \delta$. **Interpretation.** Numerically, $\delta$ acts like a conservativeness amplifier. Theoretically, it encodes a robustness margin on $\lambda_{\min}$. Mathematically, it discounts the least eigenvalue to stabilize finite-sample estimation.

**Correct scaling w.r.t. $\delta$ (using $\delta = \rho\,\lambda$).** Let $\rho \in (0,1)$ and pick an anchor $\rho_0$. Then

$$K^*(\rho) = K^*(\rho_0) \left( \frac{1-\rho_0}{1-\rho} \right)^2. \tag{16}$$

For example, from $\rho = 0.1$ to $\rho = 0.9$ the factor is $81\times$.

## G.2 Effective Rank Definition

**Result.** Rank estimation method strongly affects tightness: thresholding gives $\approx 3.4\times$, energy-based $\approx 2.4\times$, and trace-based $\approx 1.8\times$.
**Interpretation.** The definition operationalizes model complexity. Threshold is safest but loosest; energy balances dominant and tail directions; trace emphasizes overall spread.
**Mini-table.**

| Rank definition | Error ratio |
|---|---|
| Threshold | $\approx 3.4\times$ |
| Energy | $\approx 2.4\times$ |
| Trace | $\approx 1.8\times$ |

## G.3 Sub-Gaussian Parameter

**Result.** Increasing $\sigma$ from $0.5$ to $2.0$ scales $K^*$ by several times, sometimes approaching an order-of-magnitude increase.
**Interpretation.** Numerically, $\sigma$ is a dominant driver of between-task variability. Theoretically, it reflects tail decay and concentration strength. Mathematically, it is the constant in sub-Gaussian concentration appearing in upper bounds.
**Mini-table.**

| $\sigma$ | Qualitative effect on $K^*$ |
|---|---|
| 0.5 | small |
| 2.0 | several$\times$ larger |

## G.4 Covariance Norm

**Result.** Using the spectral norm typically yields $3$–$5\times$ error ratios, Frobenius yields $1.5$–$2\times$, while mean-variance approximations can under-estimate and approach $\approx 1\times$.
**Interpretation.** Spectral norm focuses on the hardest direction; Frobenius spreads mass across all directions; rough mean-variance may miss heavy directions and be overly optimistic.
**Mini-table.**

| Norm choice | Error ratio |
|---|---|
| Spectral | $3$–$5\times$ |
| Frobenius | $1.5$–$2\times$ |
| Mean-variance | $\approx 1\times$ (may under-estimate) |

## G.5 LABEL TOKENIZATION

**Result.** Single-token labels (e.g., `yes/no`, `POS/NEG`) yield smaller knees and $1.2$–$1.5\times$ error ratios, whereas multi-token labels (e.g., `positive/negative`) often exceed $3\times$.

**Interpretation.** Single-token labels shrink the effective output search space. Multi-token labels dilute probability mass and inflate the required $K$.

**Mini-table.**

| Tokenization | Error ratio |
|---|---|
| Single-token | $1.2$–$1.5\times$ |
| Multi-token | $\geq 3\times$ |

## G.6 PROMPT TRUNCATION POLICY

**Result.** *Latest-first* delivers smaller knees and $1.2$–$1.8\times$ error ratios; *earliest-first* shifts the knee right and often exceeds $3\times$; random truncation increases variance across runs.

**Interpretation.** Latest-first respects the model inductive bias toward local context; earliest-first tends to retain stale or less relevant examples; random adds sampling noise.

**Mini-table.**

| Policy | Error ratio |
|---|---|
| Latest-first | $1.2$–$1.8\times$ |
| Earliest-first | $> 3\times$ |
| Random | larger variance |

## G.7 THEORY CONSTANT SCALING

**Result.** A global scale $K_{\text{adj}}^* = C \cdot K_{\text{theory}}^*$ with $C \approx 10^{-2} \sim 10^{-1}$ systematically tightens predictions to $1.1$–$1.5\times$.

**Interpretation.** Numerically, $C$ acts as a cross-task calibrator. Theoretically, it compensates for conservative constants in concentration bounds.

**Mini-table.**

| $C$ | Typical error ratio |
|---|---|
| $10^{-2} \sim 10^{-1}$ | $1.1$–$1.5\times$ |
| $1$ | $2$–$5\times$ (uncalibrated) |

## G.8 SYNTHETIC DATA EXPERIMENTS

**Result.** Under controlled Gaussian features with known $\Sigma$, the theory-to-practice gap shrinks to $1.0$–$1.3\times$.

**Interpretation.** This indicates that the larger gaps on real data primarily arise from violations of sub-Gaussian tails, linear separability, or spectral stability.

**Mini-table.**

| Setting | Error ratio |
|---|---|
| Controlled Gaussian | $1.0$–$1.3\times$ |
| Real tasks | $2$–$5\times$ |

### G.9  RETRIEVAL STRATEGY

**Result.** Cosine-similarity nearest neighbor retrieval yields the smallest knees and $1.3$–$1.6\times$ error ratios; random sampling degrades stability and often exceeds $2.5\times$; class-balanced sampling lies between.
**Interpretation.** Retrieval reshapes the empirical covariance around the test point, preserving a larger effective $\lambda_{\min}$ and tightening $K^*$.
**Mini-table.**

| Retrieval | Error ratio |
|---|---|
| Cosine-NN | $1.3$–$1.6\times$ |
| Balanced | intermediate |
| Random | $\geq 2.5\times$ |

### G.10  MODEL SCALE

**Result.** On SST-2, the empirical knee decreases from $\approx 12$ for a 124M model to $\approx 7$ for a 345M model, with the error ratio dropping from $3.4\times$ to $2.2\times$.
**Interpretation.** Larger models better exploit demonstrations and align more closely with theoretical assumptions (smaller effective noise, more stable spectra).
**Mini-table.**

| Model | Knee (SST-2) | Error ratio |
|---|---|---|
| 124M | $\approx 12$ | $3.4\times$ |
| 345M | $\approx 7$ | $2.2\times$ |

**Closing remarks.**  Across the ten axes, three levers consistently tighten the theory-to-practice gap: (i) increasing the effective spectral gap (via $\delta$ choice, retrieval near the test point, and stable norm definitions), (ii) reducing effective noise (smaller $\sigma$, single-token labels, larger models), and (iii) cross-task calibration of conservative constants ($C$-scaling). Under idealized assumptions validated by synthetic data, the gap nearly vanishes; on real tasks, careful choices along these levers reduce error ratios toward $1.1$–$1.6\times$ while preserving reproducibility.

## H  THEORETICAL CONSTANTS: SOURCES, MAPPINGS, AND PRACTICAL CALIBRATION

### H.1  LIST OF CONSTANTS AND QUICK REFERENCE

Table 8 summarizes all explicit or implicit constants, structural factors, and engineering fudge factors appearing in the paper. The following subsections provide: (i) theoretical origins and geometric interpretations; (ii) quantitative coupling with problem parameters (dimension $d$, effective rank $r(\Sigma)$, signal-to-noise ratio, confidence level $1 - e^{-t}$); (iii) a three-step experimental calibration guideline (simulation–calibration–cross-validation).

Table 8: Constants and recommended values

| Symbol | Type | Occurrence | Recommended Range | Reference |
|---|---|---|---|---|
| $2, 2/3$ | Structural | Matrix Bernstein tail bound | Fixed, non-tunable | Tropp (2012, Thm. 1.4) |
| $c_1$ | Absolute | Fourth-moment bound | $[1.5, 6]$, recommend 3–4 | Vershynin (2018, §3.4) |
| $c_2$ | Absolute | Covariance deviation bound | $[2, 4]$, recommend 3 | Koltchinskii–Lounici (2017, Thm. 3.1) |
| $c', c_3$ | Engineering | Sample-size formulae | $\geq 8$, typically 10–15 | This work (two-stage scheme) |

## H.2 THEORETICAL ORIGINS AND GEOMETRIC INTERPRETATION

**Matrix Bernstein constants** $2$ **and** $2/3$**.**  For the sum of independent centered random matrices $S = \sum Y_k$, Tropp (2012) shows

$$\Pr(\lambda_{\min}(S) \leq -t) \;\leq\; r_{\text{eff}} \exp\left(-\frac{t^2}{2v + \frac{2}{3}Rt}\right).$$

The coefficients $2$ and $2/3$ arise from the moment-generating-function method and are structural, non-adjustable constants.

**Sub-Gaussian fourth-moment constant** $c_1$**.**  For a mean-zero sub-Gaussian vector $Z \in \mathbb{R}^d$, an absolute constant $c_1$ ensures

$$\mathbb{E}\|Z\|^4 \;\leq\; c_1 \left(\mathbb{E}\|Z\|^2\right)^2.$$

Examples:

- Gaussian: $c_1 \to 1$ as $d \to \infty$;
- Uniform on sphere: $c_1 \approx 3$;
- Rademacher: $c_1 \approx 3$.

For anisotropic distributions, $c_1$ scales with the condition number $\kappa(\Sigma)^2$. In practice, after standardization and with $d \geq 50$, simulations suggest $c_1 \in [1.5, 6]$, with 3–4 recommended.

**Covariance operator deviation constant** $c_2$**.**  Koltchinskii & Lounici (2017, Thm. 3.1) proved

$$\|\hat{\Sigma} - \Sigma\| \;\leq\; c_2 \|\Sigma\| \left(\sqrt{\frac{r(\Sigma)t}{n}} + \frac{r(\Sigma)t}{n}\right).$$

Typical values:

- Sub-Gaussian: $c_2 = 2$;
- Sub-exponential: $c_2 = 4$;
- Finite fourth moment: $c_2$ scales with kurtosis $\kappa$.

Empirically, $c_2 = 3$ suffices in 95% of cases for $d/n \in [0.1, 0.5]$.

## H.3 COUPLING WITH PROBLEM PARAMETERS

The main sample-size requirement is

$$n \;\geq\; c' \cdot r(\Sigma) \frac{t + \log d}{\varepsilon^2}.$$

Implications:

- $c_1$ influences estimation error of the effective rank $r(\Sigma)$;
- $c_2$ scales the relative error $\varepsilon$ directly;
- Structural constants $2, 2/3$ remain fixed;
- Engineering factors $c', c_3$ ensure practical safety, typically increasing with confidence level $1 - e^{-t}$.

## H.4 THREE-STEP CALIBRATION PROCEDURE

1. **Simulation calibration:** Generate synthetic Gaussian datasets across $(d, n)$ grids, estimate failure probabilities, and back out $c_2(\delta)$ with 95% quantile anchoring.

2. **Robustness to misspecification:** Test under sub-exponential, heavy-tailed, and real bootstrap data. If $c_2$ inflates more than $1.5\times$, adjust by $\kappa/5$.

3. **Cross-validation locking of $c', c_3$:** On real data, run two-stage procedures with 5-fold cross-validation, plot error–sample size curves, and infer $c'$ from the plateau point.

