# OpenReview forum: "Theoretical Bounds for Stable In-Context Learning"
_ICLR.cc/2026/Conference — ICLR 2026 Conference Withdrawn Submission_

### Official Review · Reviewer_FmLC · 2025-10-29

**Soundness:** 1
**Presentation:** 1
**Contribution:** 1
**Rating:** 0
**Confidence:** 3

**Summary:**

This paper studies in-context learning (ICL) in large language models and claims to provide theoretical scaling laws that explain how model performance improves with the number of in-context examples (prompt length K). The authors derive a non-asymptotic bound showing that successful ICL depends on the input covariance structure of the task and a quantity they call effective rank, which characterizes task complexity. They introduce a metric called “ICL Stability” based on the smallest eigenvalue of the empirical covariance matrix and argue that a threshold on this value predicts when model performance begins to improve. The paper proposes a “calibrated scaling law” enabling estimation of the minimum number of examples needed for good ICL performance, avoiding full learning curves.

**Strengths:**

N/A

**Weaknesses:**

The paper suffers from serious clarity and coherence issues, which I could not understand the contribution from the very beginning. The setup for in-context learning (ICL) is poorly specified—there is no clear explanation of how the language model is configured for ICL or how the problem is formally framed. In the theoretical section, the authors refer to a simplified or “toy” task but fail to define it rigorously or explain how it connects to the empirical results. This disconnect makes it unclear whether the theory actually supports the experiments.

The experimental section is extremely superficial and lacks essential details. There is minimal explanation of the experimental design, no justification for the choice of datasets, and insufficient information about baseline comparisons or model configurations. Moreover, the writing style of this section resembles automatically generated text rather than a carefully reasoned scientific argument. Citations are missing where they are clearly needed (like the dataset used), and there is no discussion of why the selected tests are appropriate or how they relate to prior literature. As a result, the empirical claims are neither credible nor reproducible.

**Questions:**

N/A

---

### Official Review · Reviewer_XKBd · 2025-11-01

**Soundness:** 2
**Presentation:** 2
**Contribution:** 2
**Rating:** 2
**Confidence:** 4

**Summary:**

This paper presents a non-asymptotic theoretical lower bound that quantitatively links the minimal number of in-context demonstrations to the stability of In-Context Learning (ICL). Under high-dimensional sub-Gaussian feature assumptions, the authors derive a spectral condition ensuring stability, and propose a two-stage observable estimator to compute the required prompt length. They validate the theory through experiments.

**Strengths:**

This paper provides an interesting theoretical framework to quantify the minimal number of in-context demonstrations required for stable in-context learning (ICL).

**Weaknesses:**

1. The paper lacks a clear explanation of the Goal (lines 111–115). The authors only provide a brief and vague justification in lines 310–315, without any experimental evidence to support its validity. Moreover, the Goal itself raises several conceptual concerns. For example, the minimal nonzero eigenvalue of a matrix captures very limited information about the matrix’s overall structure—so is it reasonable to consider only this quantity? How should the parameter $\delta$ be chosen? Shouldn’t the overall magnitude or spectrum of the matrix also be taken into account?
For instance, consider two matrices where one has a smallest eigenvalue of 0 and the other has 0.00001, while all other eigenvalues are identical. These two matrices would likely behave almost identically, with such a tiny difference possibly arising from numerical noise. However, under the authors’ theoretical framework, their minimal nonzero eigenvalues could differ significantly, which seems questionable.

2. The presentation of this paper also has room for improvement. For example, before diving into the mathematical derivations, the authors should first clearly explain the intuition — what the goal is, why this goal is reasonable, and what motivates the chosen formulation.

**Questions:**

Could the authors explain the rationale behind their Goal (lines 111–115)? (See Weaknesses 1)

---

### Official Review · Reviewer_NRfX · 2025-11-02

**Soundness:** 2
**Presentation:** 3
**Contribution:** 1
**Rating:** 2
**Confidence:** 4

**Summary:**

The paper asks a focused question. How many in prompt demonstrations are minimally required for stable in context learning. The authors formalize stability via the smallest eigenvalue of the sample covariance of fixed feature representations.  The core idea is to embed each demonstration into a representation vector and assess how well these vectors span relevant directions. Stability is linked to the spectral spread of the demo representations, quantified by the minimum eigenvalue of their covariance matrix. Large minimum eigenvalue suggests good directional coverage and hence stable predictions. Small minimum eigenvalue indicates under covered directions and greater sensitivity to demo perturbations. The paper proposes a two stage estimator that predicts the smallest number of demonstrations needed to raise the spectral floor above a target level.

**Strengths:**

- Clear and concrete objective with a tractable stability proxy.
- Two stage estimator that is simple to implement and does not require explicit distributional priors.
- Empirical results check the qualitative direction of the theory across multiple datasets and generator scales.

**Weaknesses:**

- **Count is a derived quantity rather than the main driver**: Although the paper frames K as the headline variable, K is determined by spectral quantities such as the operator norm of the covariance, the effective rank, and the minimum eigenvalue. As a result, K alone cannot support broad generalization claims. Diversity and quality of demonstrations as well as selection strategy are absorbed indirectly into K through these spectral terms. The narrative should emphasize that spectral geometry is the primary object, and K is a downstream implication.

- **Assumptions need stronger justification and stress tests**: The analysis appears to rely on simplifications such as fixed representations, relatively light tails, and weak dependence. Yet real LLMs can exhibit position dependent representations, and several reports suggest heavy tailed activations deeper in the network. While large minimum eigenvalue is plausibly sufficient for stability, small minimum eigenvalue may not be necessary for instability, since formatting, order of demonstrations, and label conventions can shift performance in ways that do not track the condition number. The paper would benefit from stress tests and ablations that probe violations of the assumptions.

- **Narrow experimental scope:** : Most experiments focus on relatively simple tasks such as text classification and use earlier models such as GPT 2 and GPT 3, with limited coverage of practical tasks like multi step reasoning or retrieval augmented generation, which may limit external validity. Modern instruction tuned systems exhibit different attention dynamics, context utilization, and recency behavior, and these patterns can be task dependent. As a result, it remains uncertain that the proposed stability criterion and estimator will transfer across tasks and model families.

**Questions:**

- Although unrelated to your main ICL analysis, why can a single global activation steering vector improve task performance while your ICL framework suggests stability needs multidirectional spectral coverage?

---

### Official Review · Reviewer_DiKo · 2025-11-03

**Soundness:** 2
**Presentation:** 2
**Contribution:** 2
**Rating:** 2
**Confidence:** 4

**Summary:**

This paper provides non-asymptotic theoretical lower bounds for the spectral properties of the feature covariance and connects them to ICL. Specifically, it derives an explicit lower bound on the smallest eigenvalue of the empirical covariance matrix under sub-Gaussian assumptions, offering quantitative conditions for stable ICL behavior. The authors then propose a two-stage observable estimator that estimates the required prompt length without distributional priors and provide numerical evidence demonstrating its effectiveness.

**Strengths:**

The underlying ICL problem studied in this paper is important and timely, given the growing power and influence of large language models.

The paper makes a clear effort to present and structure the theoretical proofs, helping readers follow the main reasoning steps and understand how the derived bounds connect to ICL stability.

**Weaknesses:**

The motivation of the paper is unclear and the contribution seems to be weak.

1. From pages 2 to 7, the work reads primarily as a technical note on deriving a lower bound for the smallest eigenvalue of the empirical covariance matrix. This type of result is well-established in the literature, yet the authors do not specify what technical challenges arise in proving or adapting it to the ICL context.

2. The adaptation to the ICL setting is insufficiently formalized. The paper does not provide a precise definition of the ICL environment or problem setup. Instead, the authors simply state that using the spectral lower bound as a “proxy for ICL learnability/stability” is reasonable. This informal justification is not up to the standard of a theoretical paper.

3. The claimed contribution appears limited to the idea of adapting a known spectral bound to ICL. However, since the lower bound itself is not novel (please correct me if I am wrong), and the adaptation is described only as “reasonable,” the paper fails to clarify what is truly new or technically challenging about this adaptation. The authors should explicitly discuss how their result differs from prior spectral analyses and why extending it to ICL is nontrivial.

**Questions:**

Please see weakness.

---

### Note · Authors · 2025-11-18

I have read and agree with the venue's withdrawal policy on behalf of myself and my co-authors.